# Future of Neutron Star Studies with Fast Radio Bursts

**Sergei B. Popov** [1,2,*] and **Maxim S. Pshirkov** [2,3]

[1] ICTP—International Centre for Theoretical Physics, Strada Costiera 11, I-34151 Trieste, Italy
[2] Sternberg Astronomical Institute, Universitetsky pr. 13, Moscow 119234, Russia; pshirkov@sai.msu.ru
[3] P. N. Lebedev Physical Institute of the Russian Academy of Sciences, Pushchino Radio Astronomy Observatory, Pushchino 142290, Russia
[*] Correspondence: spopov@ictp.it; Tel.: +39-040-2240-368

**Abstract:** Fast radio bursts (FRBs) were discovered only in 2007. However, the number of known events and sources of repeating bursts grows very rapidly. In the near future, the number of events will be $\gtrsim 10^4$ and the number of repeaters $\gtrsim 100$. Presently, there is a consensus that most of the sources of FRBs might be neutron stars (NSs) with large magnetic fields. These objects might have different origin as suggested by studies of their host galaxies which represent a very diverse sample: from regions of very active star formation to old globular clusters. Thus, in the following decade we expect to have a very large sample of events directly related to extragalactic magnetars of different origin. This might open new possibilities to probe various aspects of NS physics. In the review we briefly discuss the main directions of such future studies and summarize our present knowledge about FRBs and their sources.

**Keywords:** neutron stars; fast radio bursts; magnetic field; radio astronomy





## 1. Introduction

Neutron stars (NSs) are probably the most interesting physical bodies in inanimate nature, as they are very rich in extreme physical processes and conditions. These objects are far from being completely understood. Thus, any new approach to study them is welcomed, especially if it promises to provide a large new sample of sources. Observations of fast radio bursts (FRBs) is one of such examples.

The field of FRB studies was born in 2007 when the first event—the Lorimer burst—was announced [1]. FRBs are millisecond extragalactic radio transients; see a detailed review in [2]. Up to now, no counterparts in other wavelengths have been ever detected for them. An illustrative exception is the galactic source SGR 1935+2154. In April 2020, a simultaneous detection of an FRB-like radio burst [3,4] and a high-energy flare [5–8] from this magnetar happened. This resulted in the eventual that magnetars are the sources of FRBs. this does not certify that *all* FRBs are due to magnetar flares. The situation can be similar to the one with short gamma-ray bursts. Mainly, they are due to coalescence of NSs [9]. However, some fraction of events can be due to core collapse [10], some can be due to giant flares of extragalactic soft gamma-ray repeaters [11], etc. In the same way, the FRB source's population can be non-uniform, but it is widely believed now that it is dominated by magnetars [2].

Since the paper by Lorimer et al. [1] was published, many proposals to explain the origin of FRBs have been proposed (e.g., Ref. [12]). However, at the moment, the set of basic scenarios under discussion is very limited—see an extensive recent review about the most plausible emission mechanisms in, e.g., Ref. [13]—and all of them involve magnetars (see, however, Ref. [14] for descriptions of some models not involving NSs; the population of FRB sources may be non-uniform, i.e., some events can be unrelated to magnetar flares).

The idea that FRBs are related to $\gamma$/X-ray flares of magnetars was proposed already in 2007 [15]. Observations of SGR 1935+2154 basically confirm this hypothesis, but the

exact emission mechanism is still not known [16]. Presently, there are two main families of models to explain radio emission of FRBs, see, e.g., Ref. [17]. Either radio waves are produced in the magnetosphere of a magnetar, or they are due to a coherent maser-like emission mechanism operating at a relativistic shock far from the NS surface, at a typical distance $\sim 10^{14}$ cm.

Up to now, data on many hundreds ($\lesssim 10^3$) of one-off FRBs have been published (and, presumably, many more will be published soon). In addition, there are $\gtrsim 50$ repeating sources [18]. From some of them, hundreds, or even thousands in a few cases, of individual radio flares have been detected. The number of events rapidly grows with time. One-off events are actively discovered, e.g., by CHIME, the Canadian radio facility [19]. Numerous bursts from repeaters are detected due to monitoring of known sources. In particular, the FAST radio telescope [20] is very productive in this respect due to its huge collecting area.

Already, the number of sources of FRBs is by an order of magnitude comparable to the number of known radio pulsars (PSRs), see the ATNF catalog [21]. Note that the number of PSRs exceeds any other population of known sources with NSs, and the situation is not going to change qualitatively in the near future. It is expected that the square kilometer array (SKA) will discover all radio pulsars in the galaxy pointing towards us [22]. This number is just $\sim$few$\times 10^4$. On other hand, it is expected that SKA will detect $\sim 10^4$–$10^5$ FRBs per day [23]. Another proposed facility—the PUMA survey—is expected to discover $10^6$ FRBs during its operation [24]. To conclude, in the following decade, FRBs will be the most abundant sample of known NSs. It is worth noting that they are mostly going to be extragalactic sources.

Large samples of events associated with (young) strongly magnetized NSs up to redshifts $z \gtrsim$ might allow various interesting studies of NS physics.In addition, FRBs are known to be important probes of inter- and circumgalactic medium. Observations of these radio transients allow us to derive cosmological parameters and test predictions of fundamental physical theories. All these possibilities are the subject of the present review, with a focus on properties of NSs.

## 2. Different Channels for Magnetar Formation

All known galactic magnetars (see the McGill online catalog [25] at http://www.physics.mcgill.ca/~pulsar/magnetar/main.html (accessed on 19 March 2023)) are young objects whose properties (spatial distribution, association with young stellar population, etc.) indicate that mostly (or even totally) they are formed via the most standard channel— core-collapse supernovae (CCSN). Population synthesis models of the galactic magnetars usually assume that this is the only way to produce such objects. This is a valid assumption, as this channel indeed dominates over all others. Modeling shows that at least few percent of newborn NSs start their lives as magnetars [26–28]. Some studies even show that this fraction can be an order of magnitude higher [29]. For example, the rate of magnetar formation through the CCSN channel is at least once every few hundred years.

However, a highly magnetized NS can be formed via several different evolutionary channels. Below, we give the complete list:

- Core collapse;
- NS-NS coalescence;
- NS-WD coalescence;
- WD-WD coalescence;
- Accretion induced collapse (AIC).

As it is seen from the list, many channels represent evolution in a binary system. Evolution in a binary can be also important for magnetar formation via a CCSN, see, e.g., Ref. [30,31] and references therein. Importantly, an NS formed through one of these channels can belong to an old population. The problem of magnetar formation in old stellar populations is very important in FRB astrophysics in the context of host galaxies identification.

The first FRB source for which a host galaxy has been identified [32] appeared to be situated in a region of intensive star formation. The same can be said for several other sources, e.g., Ref. [33] and references therein. Still, there are also opposite examples, when an FRB source is located in a galaxy with low rate of star formation, e.g., Ref. [34] (see analysis of host galaxies of FRBs in [35]). The extreme case is FRB 20200120E situated in a globular cluster of the M81 galaxy [36], as globular clusters are known to contain only old stars. A detailed study of 23 hosts (17 for non-repeating FRBs and 6 for repeating sources) was recently presented in [37]. Contrary to some previous studies, the authors claim that FRB hosts have on average properties which a similar to majority of galaxies at corresponding redshifts. Generally, they do not find that the CCSN origin of the FRB sources contradicts observations. Still, in some peculiar cases, alternative channels might be operating.

Let us compare rates of NS formation in different channels specified above. The rate of NS–NS coalescence is about few $\times 10^{-5}$ yr$^{-1}$ as per a Milky Way-like galaxy [38]. Note that typically a NS–NS coalescence results in a black hole (BH) formation. For NS–WD coalescence, the rate is a little bit higher: $\sim 10^{-4}$ yr$^{-1}$ [39]. The coalescence of two WDs is relatively frequent $\sim 10^{-2}$ yr$^{-1}$ [40], but just a small fraction of them result in a NS formation. Therefore, WD–WD and AIC (here and below we distinguish a NS formation due to WD–WD coalescence from other types of AIC) provide the rate from few $\times 10^{-6}$ yr$^{-1}$ up to few $\times 10^{-5}$ yr$^{-1}$ [41] Here, NS formation via WD–WD coalescence does not include processes in globular clusters. About this possibility, see, e.g., Ref. [42] and references therein. Thus, altogether all channels additional to the CCSN provide less than 1% of NSs. Even if the fraction of magnetars is high in these channels, their total contribution is much less than that from the CCSN, so we expect less than one object with an age $\lesssim 10^4$ yrs per galaxy.

As the rate of NS formation due to the AIC or different coalescences is very low, it is impossible to find a representative sample of such sources in the galaxy (or even in nearby galaxies). FRB observations provide a unique possibility to probe the populations of these rare sources, even at different $z$.

At the moment, it is not known if magnetars formed through different evolutionary channels mentioned above can appear as distinguishable subclasses of FRBs sources when only radio observations are available. If it is possible in the near future to distinguish between them (at least in a statistical manner), then we have a perfect tool to study evolution of formation rates in different channels through cosmic history.

## 3. Properties of the Surrounding Medium

Pulsars have been used as excellent probes of the galactic interstellar medium (ISM) almost since their discovery. Observations of FRBs allow us to implement already developed methods to study the medium along the path from the bursts, starting from the circumburst environment and ending with the Milky Way halo and ISM; see a review in [43].

There are three major effects that affect a signal during its propagation. First, there is dispersion of the signal propagating in the plasma with electron concentration $n_e$—the group velocity $v_g$ depends on frequency $\nu$:

$$v_g = c\sqrt{1 - \left(\frac{\nu_p}{\nu}\right)^2},\tag{1}$$

where $\nu_p = \sqrt{\frac{n_e e^2}{\pi m_e}} = 8.98(n_e/1\text{ cm}^{-3})^{1/2}$ kHz is the plasma frequency, $m_e$ is the electron mass. The time delay between two observing frequencies $\nu_1$ and $\nu_2$ is :

$$\delta t \approx 4.15\text{ ms}\left[\left(\frac{\nu_1}{1\text{ GHz}}\right)^{-2} - \left(\frac{\nu_2}{1\text{ GHz}}\right)^{-2}\right]\text{DM},\tag{2}$$

where $\mathrm{DM} = \int_0^d n_{\mathrm{e}} \mathrm{d}l$ is the dispersion measure of the source. The dispersion measure is just the column density of free electrons. As a rule, DM values are given in units [pc cm$^{-3}$]; we use these units below.

In the cosmological context for sources at redshift $z$, the equation for dispersion measure is slightly modified: $\mathrm{DM} = \int_0^d n_{\mathrm{e}}/(1+z)\mathrm{d}l$. Large values of DM, strongly exceeding those expected from the galactic contribution, are the primary indicator of an extragalactic nature of the FRBs.

Second, a signal undergoes scattering during propagation through inhomogeneous medium. This results in formation of an extended exponential 'tail' in the pulse shape. Scattering is stronger at lower frequencies, $\tau \propto \nu^{-\alpha}$, with the spectral index $\alpha$ which depends on properties of inhomogeneities. For the Kolmogorov spectrum of inhomogeneities, $\alpha = 4.4$.

Third, in a presence of magnetic fields, the polarization position angle of a linearly polarized signal would experience frequency (or wavelength)-dependent Faraday rotation:

$$\Delta \Psi = \mathrm{RM}\, \lambda^2, \tag{3}$$

where $\lambda$ is the wavelength and RM is the rotation measure:

$$\mathrm{RM} = \frac{e^3}{2\pi m_{\mathrm{e}}^2 c^4} \int_0^d n_{\mathrm{e}} B_{||} \mathrm{d}l = 812 \int_0^d n_e B_{||} \mathrm{d}l \ \mathrm{rad}\ \mathrm{m}^{-2}. \tag{4}$$

Here $n_{\mathrm{e}}$ is measured in cm$^{-3}$, $B_{||}$ is the component of the magnetic field measured in μG (positive when directed towards the observer) parallel to the line of sight, and all distances are measured in kpc. If rotation takes place at redshift $z$ there would be a correction: $\mathrm{RM}_{\mathrm{obs}} = \mathrm{RM}_{\mathrm{int}}/(1+z)^2$, i.e., the observed $\mathrm{RM}_{\mathrm{obs}}$ would be smaller than the intrinsic $\mathrm{RM}_{\mathrm{int}}$.

All plasma along the path contributes to these effects: there are contributions from the host galaxy (including a circumburst region), the intergalactic medium (IGM), and the halo and the ISM of the Milky Way. For some bursts, there would be considerable contribution from circumgalactic medium of intervening galaxies located at the line of sight and from regions of the large-scale structure, such as galaxy clusters and filaments.

The relative contributions from these regions are different for the three effects mentioned above. While DM is mostly accumulated in the IGM, most of the scattering comes from the ISM in the host galaxy and the Milky Way. Finally, the Faraday rotation mostly takes place in the ISM of galaxies and, especially, in the circumburst medium (CBM).

NSs born via various evolutionary channels discussed in the previous section might have different properties of the surrounding medium. Some valuable information could be extracted from the observations of one-off bursts, e.g., detection of excessive scattering, which is most probably associated with the CBM, might shed light on properties of turbulence in the immediate vicinity of some bursts [44]. Still, observations of repeating bursts are better suited for studying the CBM. Recurring activity gives us an opportunity to use variety of instruments, working with different temporal resolutions and in different frequency ranges. For example, emission from FRB 121102 initially has been supposed to be unpolarized. Only subsequent follow-up observations of this repeating source at higher frequencies with high temporal resolution let the authors measure the degree of linear polarization and to obtain an extreme value of the rotation measure: $\mathrm{RM} \sim 10^5$ rad m$^{-2}$ [45].

Even more importantly, observations spanning several years could give an opportunity to detect time evolution of DM and RM, therefore seriously constraining properties of the CBM because the evolution of the CBM would be the leading factor in the observed variation of DM and RM [46–48]. In the early stage of expansion of a SNR, likely the most relevant situation for repeaters, DM evolves as $t^{-1/2}$ if the supernova exploded into the medium of constant density, and DM $\propto t^{-3/2}$ if the explosion took place in a wind-formed environment. For RM, the scaling in such a situation is $t^{-1/2}$ and $t^{-2}$, correspondingly [47].

RM evolution was relatively quickly discovered in the case of FRB 121102. Two and a half years of observation demonstrated the rapid decrease of RM in the source frame from $1.4 \times 10^5$ rad m$^{-2}$ to $1.0 \times 10^5$ rad m$^{-2}$ just in one year and considerable leveling-off afterwards. The DM slightly *increased* by $\sim 1$ pc cm$^{-3}$. This behavior could be explained by the evolution of a very young pulsar wind nebula (PWN) embedded in a supernova remnant (SNR) with an age of about 10–20 years.

An even more extreme example was presented by observations of FRB 190520B. This burst has one of the largest excess DMs known, $\sim 900$ pc cm$^{-3}$, which is decreasing at an astounding rate of $\sim 0.1$ pc cm$^{-3}$ day$^{-1}$. Thus, the inferred characteristic age is only 20–30 years. Between June 2021 and January 2022, the observed RM demonstrated an extreme variation from $\sim +10{,}000$ rad m$^{-2}$ to $\sim -16{,}000$ rad m$^{-2}$, implying a drastic reversal of the *B*-field of $\sim$ mG strength. This behavior can again be explained by a SNR evolution or by a close proximity to a massive BH with strongly magnetized outflows, or alternatively by a magnetized companion [49]. In the latter case, the RM and DM variations could be periodic, and this will be tested in the near future. It could be a relevant fact that FRB 121102 and FRB 190520B are the only bursts known which have spatially coinciding persistent radio sources. These sources could be related to regions which produce extreme behavior of the RM. Some other repeaters also demonstrate RM variations, although these are not so extreme [50]. Extensive study of the varying magneto-ionic environment of 12 repeating FRBs was performed in [51], where it was shown that the RM variations in these FRBs are much more extreme than in known young pulsars in the Milky Way. This may imply that the properties of the surrounding medium are considerably different in these two cases.

It is obvious that detection of many more new repeaters in a wide range of NS ages would significantly expand our understanding of the earliest epoch of evolution of complicated systems comprising NS: PWN and SNR. Another way to study the CBM, or at least to constrain models which suggest interaction of magnetar flares with the surrounding medium as a source of FRBs, is to search for the prompt emission and an afterglow from FRBs at different frequencies, including optics and X-rays [52–54]. Due to the weakness of the expected signal, future observations of the galactic (SGR 1935+2154) or nearby extragalactic FRBs would be especially valuable.

## 4. Very Short-Term Periodicity and Quasiperiodic Features

Up to now, there have been no robust measurements of spin periods of the sources of FRBs (see also the next section). Still, there are already several very interesting and important results related to short-term (quasi)periodicity.

Firstly, this is the periodicity detected (at the $6.5\sigma$ significance level) in a burst of the one-off source FRB 20191221A [55]. The event is atypically long—$\sim 3$ s—and has a complicated structure. At least nine components are well distinguished. Analysis demonstrates that these components are separated by intervals which are multiples of 0.217 s (no significant deviations from the strict periodicity in the time of arrivals of single components are observed). The origin of this periodicity is unclear.

It is tempting to say that the periodicity reflects the spin of the NS. This can be checked if another burst with periodicity is detected from this source. If we are dealing with a young magnetar with the spin period 0.217 s, then it must have $\dot{P} \sim 10^{-9}$–$10^{-8}$. On a time scale of several years it might be quite easy to detect its spin-down.

Another possibility discussed in [55] is related to quasiperiodic oscillations similar to those detected in galactic magnetar flares (e.g., SGR 1806-20 [56] and SGR J15050-5418 [57]). They have frequencies from $\sim 20$ Hz up to $\sim 1$ kHz, i.e., somewhat higher than in the burst of FRB 20191221A. Alternatively, periodicity can be related to magnetospheric processes. However, this possibility is less probable (see discussion in [55]).

Quasiperiodic behaviour with the frequency $\sim 40$ Hz is also suspected for one burst of the galactic magnetar SGR 1935+2154 [58]. This is exactly the burst which was observed simultaneously in radio and gamma/X-rays. The quasiperiodic structure is found in the

high-energy data obtained by *Insight*-HXMT. The identification of this quasiperiodicity is not very significant (3.4 $\sigma$) as only three peaks are well identified in the burst. However, the result is very intriguing. New observations of this source might clarify the situation.

Let us move towards higher temporal frequencies. Observations of FRB 20201020A demonstrated the existence of a quasiperiodic structure with characteristic time scale 0.415 msec [59]. In this one-off FRB, observations with Apertif distinguished five components.

Of course, a submillisecond spin period can be excluded, as the frequency is very high. It is even too high for crustal oscillations. A frequency $\sim$2 kHz can appear in a NS–NS coalescence, and now there is an observational example: quasiperiodic oscillations are discovered in two short GRBs [60]. Still, this possibility looks rather exotic. It is more probable that the 0.415 msec structure is due to properties of a magnetospheric emission mechanism; see discussion in [59]. If so, more data on such features in FRB emission will open the possibility to study in more detail the emission properties of magnetospheres of extreme magnetars.

In radio observations, even nanosecond time scales can be probed. This opens a possibility to study processes in magnetospheres of the sources or/and at relativistic shocks (depending on the emission mechanism), as well as vibrations of a NS crust.

At the moment, the resolution$\sim$tens of nanoseconds has already been reached in FRB observations. In one case (repeating FRB 20200120E associated with a globular cluster of the M81 galaxy) it was demonstrated that sub-bursts are structured at the $\sim$2 $\mu$sec scale [61]. In this case, the feature is most probably related to a magnetospheric emission mechanism. In the case of the Crab pulsar, observations show the existence of pulses with duration <1 nanosecond [62]. These events definitely have magnetospheric origin, but the exact nature remains unclear.

Observations of FRBs might open a wide perspective of studies of periodic and quasiperiodic processes related to different aspects of NSs physics (crust oscillations, magnetospheric processes, etc.). Accounting for the growing number of observations of repeating and one-off sources at different frequencies, in the near future, this might be an important channel of information about magnetars. Still, it is also very important to determine the basic temporal characteristic of a NS: its spin period.

## 5. Spin Periods

Measurements of the spin period and its derivative, $\dot{P}$, of the first radio pulsar made it possible to identify the nature of the emitting object [63]. Spin measurements for sources of FRBs are very much welcomed, as they can allow us to understand better the properties of these NSs, in particular to prove their magnetar nature.

Determination of a spin period can give some clues to the magnetar properties. If the period derivative is measured, too, then it is possible to estimate the characteristic age $\tau_{\rm ch} = P/2\dot{P}$ and the magnetic field of the NS with the simplified magneto-dipole formula:

$$I\omega\dot{\omega} = \frac{2\mu^2\omega^4}{3c^3}. \tag{5}$$

Here $I$ is the moment of inertia of a NS, $\omega = 2\pi/P$—spin frequency, $\mu$—magnetic moment, and $c$—the speed of light. Such measurements might help us to understand the origin of the source.

Short spin periods will definitely point towards young ages of the magnetars, as $\tau_{\rm ch} \propto P^2/B^2$ if the initial spin period, $P_0$, is much smaller than the observed one. Long spins can be explained in several models (see, e.g., Ref. [64] and references therein in application to the 1000-s pulsar GLEAM-X J162759.5-523504.3).

One option is related to the fallback accretion soon after the NS formation [65]. Fallback matter can form a disc around a newborn NS. Interaction of a magnetar with the fallback disc can result in significant spin-down, e.g., Ref. [66] where the authors explain the 6.7 h period of the NS in the supernova remnant (SNR) RCW 103 and [67] (see also the next section). Another option is related to a large initial field which at some point stops to decay

significantly, so the NS rapidly spins down: $P \propto \mu \sqrt{t} \approx 10\,\mathrm{s}\,\mu_{33}(t/3000\,\mathrm{yrs})^{1/2}$ for $P \gg P_0$. Here $\mu_{33} = \frac{\mu}{10^{33}\,\mathrm{G\,cm}^3}$.

Spin periods can be measured directly or indirectly by different observations and analysis. Some (quasi)periodic structures in bursts (see the previous section) can provide information about the spin. Alternatively, appearance of repeaters' bursts can be phase dependent. From several repeating sources many hundreds of bursts are detected, see analysis in [68]. Potentially, such huge statistics can provide an opportunity to search for the spin period.

Unfortunately, there is little hope to obtain a period value using large statistics of the burst of repeating sources. It can be understood if we consider that FRB bursts might be related to high energy flares of magnetars. It is well known that some galactic magnetars produced hundreds of detected high-energy flares. However, even with such significant statistics, their distribution along the phase of spin period is typically found to be consistent with the uniform distribution, see, e.g., Refs. [69,70]. The reverse task—determination of the spin period of a soft gamma-ray repeater from burst timing statistics—cannot be performed. The same might be true for FRBs, especially if radio emission is produced far away from a NS in a relativistic shock. However, the analysis of numerous bursts from two very active repeaters gives us an opportunity to find a different type of periodicity, which we discuss in the following section.

## 6. Long-Term Periodicity

The source FRB 180916.J0158+65 (aka R3) is an active nearby repeater situated in a star-forming region in a spiral galaxy at 149 Mpc from the Earth. Relative proximity and high rate of bursts resulted in many observational campaigns dedicated to this particular source. CHIME observations resulted in a discovery of 16.5-day periodicity in activity of this source [71]. Later on, observations with different instruments at different frequencies confirmed this result.

The 16.5-day cycle consists of a (frequency dependent) window of activity and a quiescent period. Immediately, several different interpretations of the observed periodicity were proposed. Below we briefly describe three models proposed in the magnetar framework. Still, alternative explanations are also possible; for example, a model based on an accretion disc precession is described in [72].

Probably, the most natural assumption which can explain the detected long-term periodicity is the binarity of the source [73], see also [74] for a review and development of the model. Time scale $\sim$10–20-days is quite typical for orbital periods of binary systems with a NS and a massive companion, see a catalog of high-mass X-ray binaries in [75]. Intensive stellar wind from the massive star (e.g., a supergiant) would provide an environment which, for example, can modulate (frequency dependent) windows of transparency for the radio emission of the NS. It is not difficult to formulate a realistic scenario of formation of a magnetar in such systems [76].

Another option is related to precession. Understanding free precession of NSs is a long-standing problem [77,78]. A NS might not be an ideally symmetric object, but oblate (biaxial in the first approximation), with non-equal principal moments of inertia $I_3 > I_2 = I_1$. If $I_3 = I_1(1 + \epsilon)$ where $\epsilon = (I_3 - I_1)/I_1 \ll 1$ is the oblateness, then, the precession period can be written as $P_\mathrm{p} \approx P/\epsilon$. For $P \sim 1\,\mathrm{s}$ and $\epsilon \sim 10^{-6}$ we obtain a precession period similar to the one observed for the R3.

Finally, the third proposed hypothesis simply relates the observed periodicity to the spin period of the NS [79]. As was already mentioned in the previous section, there are several variants regarding how a NS can achieve a very long spin period—much larger than observed for the vast majority of PSRs or/and known galactic magnetars.

R3 is not the only source of FRBs for which that type of periodicity is detected. The first repeater—FRB 121102—became the second one for which activity is limited to periodically repeating cycles. In the case of FRB 121102, the period appeared to be an order of magnitude longer [80,81].

Up to now, all three scenarios to explain the observed periodicity seem to be plausible. No doubt, in the near future, the same type of behavior will be discovered for other sources. In any case, this will open opportunities to obtain new information about NSs. This is very promising, because up to now we were unaware of active magnetars in binaries [82] or precessing magnetars as well as NSs with spin periods ∼10–100 days. Therefore, regardless of which option is correct, a growing sample of FRB sources with periodic activity will bring us new information about the physics and astrophysics of NSs.

However, observations of FRBs can be useful not only for NS studies, but also for testing the fundamental properties of nature and measuring some basic physical parameters.

## 7. Fundamental Theories

FRBs in many respects are unique sources. They produce very narrow bursts (sometimes with microstructure visible down to the scale of tens of nanoseconds) and they are visible from cosmological distances corresponding to $z > 1$. This makes them a powerful tool to measure (or put limits on) some fundamental parameters.

### 7.1. Testing the Equivalence Principle

In general relativity (GR), photons of different energies experience the same gravitational effects. For example, if a burst with extended spectrum is emitted at a cosmological distance, then we expect to receive all photons at the same time (neglecting dispersion of the signal in the medium). However, in many theories of gravity, this is not the case. Thus, astronomical observations of distant transient sources can be used to test theoretical predictions.

Historically, gamma-ray bursts (GRBs) were actively used for fundamental theories tests, e.g., Ref. [83] and references therein. However, FRBs have some advantages due to their very sharp short pulses and the high precision of radio astronomical observations. These sources were proposed as probes for the Einstein equivalence principle soon after their discovery [84].

Testing the equivalence principle is typically defined as a limit on the post-Newtonian parameter $\gamma$. This quantity defines how much curvature is produced by unit rest mass. In some cases it can be given as:

$$\gamma = \frac{1 + \omega_{\mathrm{BD}}}{2 + \omega_{\mathrm{BD}}}, \tag{6}$$

here $\omega_{\mathrm{BD}}$ is the Brans–Dicke dimensionless parameter. GR is reproduced for $\omega_{\mathrm{BD}} \to \infty$ (i.e., if $\gamma = 1$).

Observationally, the delay $\Delta t$ between signals at different frequency is measured. The hypothetical effect of the equivalence principle violation can be hidden by others, but if we separate it then we obtain:

$$\Delta t = \frac{\gamma(\nu_1)}{c^3} \int_{r_{\mathrm{em}}}^{r_{\mathrm{obs}}} U(r)\mathrm{d}r - \frac{\gamma(\nu_2)}{c^3} \int_{r_{\mathrm{em}}}^{r_{\mathrm{obs}}} U(r)\mathrm{d}r. \tag{7}$$

Here, $\nu_1$ and $\nu_2$ are two different frequencies of electromagnetic radiation. $U(r)$ is the gravitational potential. The integral is taken from the point of emission to the point of detection.

A simplified conservative approach is based on the time delay due to photon's propagation in the gravitational potential of the galaxy in Equation (7), e.g., Ref. [84]. Such an approach results in the limits $\Delta\gamma \equiv |\gamma - 1| \lesssim 10^{-8}$. However, detailed calculations for cosmological sources are non-trivial [85]. Calculations along the cosmological path requires an accurate consideration, as just decaying continuation of the galactic potential in Minkowski space leads to an incorrect result. Under reasonable assumptions about the value of the effect during the whole trajectory on a cosmological scale, much more tight limits can be derived. For example, in [86], the authors obtain $\Delta\gamma \lesssim 10^{-21}$ for FRBs beyond $z = 1$.

Most probably, in the near future FRB observations will remain the most powerful astronomical tool to test the equivalence principle. This might be possible not only due

to an increase of the number of known sources, discovery of bursts at a larger redshift, and improvements in the model parameters, but also due to usage of new measurements, e.g., related to statistical properties of the dispersion measure [87].

In principle, the violation of the Lorentz-invariance can also be tested with FRBs, especially if gamma-ray counterparts are detected. Such observations for extragalactic FRBs will be quite possible in the near future as FRBs are already detected at distances of about a few Mpc and gamma-detectors can detect a hyperflare of a magnetar at distances about few tens of Mpc, e.g., Ref. [88] and references therein.

*7.2. Measuring the Photon Mass Limits*

Another fundamental parameter which can be constrained by FRBs observations is the photon mass, $m_\gamma$. If photons have non-zero masses then the velocity of their propagation becomes frequency-dependent:

$$v = c\sqrt{1 - \frac{m_\gamma^2 c^4}{E^2}} \approx c\left(1 - \frac{1}{2}A\nu^{-2}\right). \tag{8}$$

Here $m_\gamma$ is the photon mass and $E$—its energy; $A = \frac{m_\gamma^2 c^4}{h^2}$.

Different methods are used to put a limit on $m_\gamma$. In particular, astronomical rapid transient sources at cosmological distances can be a very good probe. Previously, the most strict limit on the photon mass derived with such sources (GRBs) was $m_\gamma \lesssim 10^{-43}$ g. With FRBs, it became possible to improve it significantly.

If we observe a source at a redshift $z$, then the time delay between two simultaneously emitted photons with frequencies $\nu_1$ and $\nu_2$ due to a non-zero photon mass can be written as:

$$\Delta t_{\mathrm{m}} = \frac{A}{2H_0}\left(\nu_1^{-2} - \nu_2^{-2}\right)H_1(z). \tag{9}$$

Here $H_0$ is the present day Hubble constant and $H_1$ is defined as:

$$H_1 = \int_0^z \frac{(1+z')^{-2}\mathrm{d}z'}{\sqrt{\Omega_{\mathrm{m}}(1+z')^3 + \Omega_\Lambda}}. \tag{10}$$

Thus, if the redshift of an FRB source is known, then timing information about properties of the burst (width of pulses, distance between subpulses) can be used to constrain $m_\gamma$. Usage of FRBs to constrain the photon mass was proposed independently in two papers [89,90]. Curiously, these authors based their estimates on an erroneous identification of FRB 150418 with a galaxy at $z \approx 0.5$. The derived limit was $m_\gamma \lesssim 3 \times 10^{-47}$ g.

The first secure identification of the host galaxy of FRB 121102 was made a year later [32]. Immediately, this information was used [91] to put a realistic limit on the photon mass. The value appeared to be of the same order: $3.9 \times 10^{-47}$ g. Note that this limit is much better than those obtained with GRB observations.

In [92] the authors also used observations of FRB 121102 and slightly improved the limit as they used a distance between well-measured subpulses: $m_\gamma \lesssim 5.1 \times 10^{-48}$ g. Later, in [93], the authors used nine FRBs with known redshifts to put a joint limit on $m_\gamma < 7.1 \times 10^{-48}$ g. Finally, the authors of [94] obtained a better limit on the basis of the data on 17 well-localized FRBs: $m_\gamma < 4.8 \times 10^{-48}$ g.

Future simultaneous observations at significantly different frequencies of very narrow pulses with nanosecond scale time resolution will help to improve the limits on $m_\gamma$ significantly.

## 8. Discussion

*8.1. Intergalactic Medium and Baryonic Matter*

FRB observations is a very powerful tool to study the medium along the propagation from the FRB source to the Earth. It is particularly suited for studies of the IGM.

The analysis of DMs of localized FRBs (i.e., with measured $z$) could be used to search for so-called "missing baryons". Various cosmological probes show that the baryons make up around 5% of the total energy density of the Universe [95]. Still, only a minor fraction of these baryons was detected in observations of galaxies and galaxy clusters. The most popular explanation is that the remaining baryons reside in the IGM, and due to its tenacity are almost undetectable by direct observations. However, DM measurements produce the total column density, thus they are ideally suited for the task.

For this test, one needs to extract the IGM-related part $DM_{IGM}$ from the total DM, which is the sum of several components:

$$DM = DM_{MW,ISM} + DM_{MW,halo} + DM_{intervening} + DM_{IGM} + DM_{host}, \quad (11)$$

where the first and the second terms describe contributions from the Milky Way (MW) ISM and the halo, correspondingly. $DM_{host}$ combine contributions from the halo and the ISM of the host galaxy, including the circumburst region. For the aims of this analysis it is better to avoid bursts where there are intervening galaxies with large $DM_{intervening}$ close to the line of sight.

$DM_{MW,ISM}$ could be estimated using existing models of electron distribution in the Galaxy [96,97] with precision around 20%. The MW halo contribution is usually assumed to be less than $100 \text{ pc cm}^{-3}$ and a benchmark value $DM_{MW,halo} = 50 \text{ pc cm}^{-3}$ is frequently used [98]. It is crucial to estimate the host contribution, which is also around $O(100 \text{ pc cm}^{-3})$. At the moment these estimations are performed using statistical distributions [98,99], informed mainly by the cosmological simulations. The host contribution is modeled using the log-normal distribution with a median $\exp(\mu)$ and a logarithmic width parameter $\sigma_{host}$. The parameter space is studied in a wide range, e.g., in [98] $\mu$ was set in the range 20–200 $\text{pc cm}^{-3}$, $\sigma_{host}$ in the range 0.2–2.0. Host contribution parameters are included in the joint fit, along with the baryon fraction $\Omega_b$. The analysis in [98] shows that this distribution with parameter values of $\mu = 100 \text{ pc cm}^{-3}$ and $\sigma_{host} \sim 1$ successfully describes the observations. The host contribution comprises contributions from the host galaxy and the CBM. Although the latter one could be large for very young sources, it becomes subdominant ($<100 \text{ pc cm}^{-3}$) after several decades of the evolution of the remnant [47] thus it would not affect the analysis of the majority of one-off bursts. An expected increase in quality of observations and modelling of host galaxies and the CBM evolution will certainly increase the precision of $DM_{host}$ estimates.

Finally, individual realizations of $DM_{IGM} \equiv DM - (DM_{MW,ISM} + DM_{MW,halo} + DM_{host})$ are prone to inevitable fluctuations due to inhomogeneities in the IGM, so it is necessary to work with averaged (binned) values $\overline{DM}_{IGM}(z)$.

This observational estimate should be compared with the theoretical expectations (e.g., Ref. [100]):

$$DM(z) = \frac{3cH_0\Omega_b}{8\pi G m_p} \int_0^z \frac{(1+z')f_{IGM}(z')\chi(z')}{\sqrt{(1+z')^3\Omega_m + \Omega_\Lambda}}dz', \quad (12)$$

where $m_p$ is the mass of the proton, $f_{IGM}$ is the fraction of baryons residing in IGM, given that some baryons are sequestered in the stars, stellar remnants, ISM in galaxies, and so on. $\chi(z)$ is the number of free electrons per one baryon and it depends on ionization fractions $\chi_H(z)$ and $\chi_{He}(z)$ of hydrogen and helium respectively:

$$\chi(z) = Y_H\chi_H(z) + Y_{He}\chi_{He}(z), \quad (13)$$

$Y_H = 3/4$, $Y_{He} = 1/4$ are mass fractions of hydrogen and helium. For $z < 3$ both species are fully ionized, so $\chi(z < 3) = 7/8$.

As there is a degeneracy between $f_{IGM}$ and $\Omega_b$ there are two ways to exploit DM data. First, one can fix $f_{IGM}$ from some models of galaxy evolution and put constraints on $\Omega_b$: the latest results from observation of 22 localized bursts gave stringent constraints: $\Omega_b = 0.049^{+0.0036}_{-0.0033}$ [101]. Alternatively, the $\Omega_b$ value could be fixed using, e.g., cosmic microwave background (CMB) observations and some meaningful constraints on the $f_{IGM}$

could be obtained: $f_{\text{IGM}} = 0.927 \pm 0.075$ [99]. In any case, it could be stated that the long-standing problem of 'missing baryons' has been solved using FRB observations. At the moment, estimates of $f_{\text{IGM}}$ due to limited statistics found no evidence of redshift evolution. However, such evolution is expected: at higher redshifts, the fraction of baryons residing in IGM increases, approaching unity at $z > 5$. Simulations show that $N = 10^3$ of localized FRBs would be enough to detect this evolution at statistically significant level and begin to probe various models of accretion of matter from IGM [102].

Furthermore, as can readily be seen from Equation (12), DM observations could be used to study the reionization history of the universe given by the function $\chi(z)$. Robust detection of He II reionization, which is expected to occur at $z \sim 3$, could be achieved with detection of $N = 500$ FRBs with $z < 5$. The moment of sudden reionization would be pinpointed with $\delta z = 0.3$ precision [103]. Even more ambitious goals could be reached if FRB were produced by remnants of Pop III stars at $z = 15$ onwards. $N = 100$ of localized FRBs with $5 < z < 15$ redshifts could constrain CMB optical depth at $\sim 10\%$ level. This might let us find the midpoint of reionization with 4% precision; detection of $N = 1000$ FRBs would give an opportunity to describe the whole history of reionization [104]. It also would be very important for CMB analysis, reducing uncertainties on the CMB optical depth due to reionization and leading to more precise determination of various cosmological parameters, such as the amplitude of the power spectrum of primordial fluctuations.

Properties of halos of intervening galaxies could also be inferred from DM observations: only $N = 100$ of localized FRBs at $z < 1$ would suffice to mildly constrain the radial profile of CGM [105]. With $N = 1000$, it would be possible to describe this profile much better and for different types of galaxies.

Scattering mostly arises in the ISM of the host galaxy and the Milky Way. After extraction of the first contribution it would be possible to thoroughly study turbulence of the ISM for a very large set of galaxies at different redshifts [106].

The largest contribution to Faraday rotation also comes from the ISM of the galaxies, including circumburst medium. That makes them less suitable for studies of the intergalactic magnetic fields (IGMF). Still, as there is strong cosmological dilution of observed RMs: $\text{RM}_{\text{obs}} = \text{RM}_{\text{int}}/(1+z)^2$, observations of 100–1000 distant enough FRBs with $z > 2$–3 could be used to study origin of magnetic fields and discriminate between their primordial and astrophysical origin [107,108]. However, this requires extremely precise knowledge of the MW contribution at 1 rad m$^{-2}$ level. Non-trivial limits on the IGMF with very small correlation lengths ($<$kpc), which are extremely difficult to constrain by other means, were recently obtained by analysis of FRB scattering; the presence of $\mathcal{O}(10 \text{ nG})$ fields shift the inner scale of turbulence, boosting the scattering [109].

Naturally, observations of FRB RMs could be used to study magnetic fields in the host galaxies. Existing observations already gave an opportunity to detect magnetic fields with average magnitude $\left\langle B_{||} \right\rangle \sim 0.5$ µG in nine star-forming galaxies at $z < 0.5$ [110]. In the future, it will be possible to investigate magnetic fields in hundreds and possibly thousands of galaxies of different types.

### 8.2. Gravitational Lensing of FRBs

Their high-rate and short burst duration make FRBs very attractive candidates for gravitational lensing studies. Gravitational lensing is caused by the deflection of electromagnetic waves by a massive body (lens) located very close to the line of sight towards the source. In the simplest case of the point-like mass (which is true, e.g., in the case of primordial BHs), the characteristic angular scale is set by the so-called Einstein radius:

$$\theta_{\text{E}} = 2\sqrt{\frac{GM_{\text{l}}}{c^2} \frac{D_{\text{ls}}}{D_{\text{s}} D_{\text{l}}}}, \tag{14}$$

where $M_{\text{l}}$ is the lens mass, $D_{\text{ls}}$, $D_{\text{l}}$, $D_{\text{s}}$ are the distances from the lens to the source and from the observer to the lens and to the source, correspondingly.

Gravitational lensing by a point-like lens will produce two images, with the following angular positions:

$$\theta_{1,2} = \frac{\beta \pm \sqrt{\beta^2 + 4\theta_{\rm E}^2}}{2}, \tag{15}$$

where $\beta$ is the impact parameter: the angular distance between the unperturbed position of the source and the lens. In case of FRBs, we are mostly interested in the temporal properties and we would talk about two bursts, rather than two images [111].

It takes a different time for the signal to travel by two slightly different trajectories, so a certain time delay would emerge between these bursts:

$$\Delta t = \frac{4GM_{\rm l}}{c^3}(1 + z_{\rm l})\left[\frac{y}{2}\sqrt{y^2 + 4} + \ln\left(\frac{\sqrt{y^2 + 4} + y}{\sqrt{y^2 + 4} - y}\right)\right], \tag{16}$$

where $y \equiv \beta/\theta_{\rm E}$ is the normalized impact parameter, $z_{\rm l}$ is the lens redshift.

Another important property is the relative brightness of two bursts:

$$R = \frac{y^2 + 2 + y\sqrt{y^2 + 4}}{y^2 + 2 - y\sqrt{y^2 + 4}} > 1, \tag{17}$$

this means that the first burst is always brighter than the second one. In all other respects, the properties of two bursts might be very close, besides some minor differences caused by propagation in a medium with slightly different properties.

From Equation (16), it can be seen that the delay time is several times larger than the time of crossing of the gravitational radius of the lens, i.e., the $\mathcal{O}({\rm ms})$ delay corresponds to a $\sim 30\ M_\odot$ lens. Note, that here we do not discuss limitations related to scattering, see [112].

Compact objects such as primordial black holes (PBHs) are natural targets for searches with FRB lensing [113]. Initial searches were performed with the shortest detectable delay corresponding to duration of burst and succeeded in constraining the PBH fraction in the dark matter $f_{\rm PBH} < 1$ ($f_{\rm PBH} \equiv \Omega_{\rm PBH}/\Omega_{\rm DM}$) for $M_{\rm PBH} > 30\ M_\odot$. Recently, a new approach was developed: instead of correlating intensity curves it was suggested to look for correlation in the voltage (or, equivalently, electric field) curves. That gave an opportunity to progress from incoherent to coherent methods with the lower limit on the detectable time delay now set by the Nyquist frequency, $\mathcal{O}({\rm ns})$ and, correspondingly, sensitive to the lenses in the mass range $10^{-4}$–$10^4\ M_\odot$ [111,114].

What are the prospects of this method? Using the word pun from [115] they are "stellar": with $5 \times 10^4$ FRBs detected during several years of operations of next generation instruments, it would be possible to constrain PBHs fraction at $<10^{-3}$ level in the whole $10^{-4} - 10^4\ M_\odot$ mass range, setting the most stringent limits there.

FRBs could also be lensed by much more massive objects, such as galaxies. In this case, the corresponding time delay $t_{\rm d}$ will be $\mathcal{O}(10\ {\rm days})$. In this case, the best strategy would be to search for lensed repeating bursts. The short duration of bursts would make it possible to determine $t_{\rm d}$ with extreme precision, much higher than allowed by observation of lensed AGNs, where an error in $t_{\rm d}$ can exceed several hours. Measurement of time delays, along with the gravitational model of the lens (galaxy), allow one to estimate the Hubble constant and curvature term $\Omega_{\rm k}$ in a straightforward and cosmological model-independent way. Observations of 10 lensed repeaters would give an opportunity to constrain $H_0$ at sub-percent level and reach $<10\%$ precision for $\Omega_{\rm k}$ determination [116]. However, given that the lensing probability is around $3 \times 10^{-4}$ and repeaters fraction is $\sim 3 \times 10^{-2}$ that would demand $\mathcal{O}(10^6)$ detected FRBs or, possibly, several decades of SKA operations. More optimistically, almost the same level of precision, $\frac{\Delta H_0}{H_0} \sim 10^{-2}$, $\frac{\Delta\Omega_{\rm k}}{\Omega_{\rm k}} \sim 10^{-1}$ could be reached with 10 lensed non-repeating FRBs [117], which decreases the number of needed detections to $\sim$30,000.

## 9. Conclusions

FRB study is a new frontier of NS astrophysics. If we talk about magnetar bursting activity, then the statistics of FRBs are already larger than the statistics of SGR flares.If we consider just absolute numbers of known NSs related to any kind of astrophysical source, then statistics of FRBs are already comparable with the PSR statistics, and will soon significantly outnumber them.

FRB studies initiated advances in methods of radio observations of short transients. New instruments are under construction or under development. This promises new discoveries. Already, studies suggest that the known population of FRBs can be supplemented by shorter and longer events. In [118] the authors used Parkes archive of low-frequency observations to look for new FRBs. Indeed, they found four events which have specific features: a duration of $\gtrsim 100$ ms, i.e., they are longer than typical FRBs by more than an order of magnitude. The authors suggest that such long events are often missed in standard FRB searches. The same situation can occur with very short events. In [119], the authors presented the discovery of FRB 20191107B with an intrinsic width 11.3 µs. Observations have been performed with the UTMOST radio telescope. The authors argue that such short bursts can be mostly missed by UTMOST and in many other surveys. In future, observations of FRBs with non-typical (from the present point of view) properties can bring new information and new puzzles.

Intense observations can result in the discovery of new types of radio transients which are not related to FRBs. Since 2007 (and especially since 2013) numerous interesting theoretical models have been proposed in order to explain FRBs [12,120]. However, now we can treat many of them as predictions for new types of events. Thus, predictions of radio transients from cosmic strings [121], PBHs evaporation [122], white holes [123], deconfinement [124], etc., can be verified.

Deeper understanding of the physics of FRB emission and related processes together with better knowledge of the astrophysical picture of the FRB sources' formation and evolution will allow obtaining even more information using FRBs as probes and indicators. The reason is in the better understanding of links and correlations of different observables with physical and astrophysical parameters. If we understand the emission mechanism, the origin of different types of (quasi)periodicities, polarization properties, etc.—then we can directly calculate related physical parameters of NSs, and so with a large sample of observed bursts, we can obtain statistically significant information about these properties.

For example, FRBs can be related to glitches (and/or anti-glitches) of NSs. This possibility is based on recent observations of the galactic magnetar SGR 1935+2154. A few days before the famous FRB-like burst (28 April 2020) a strong glitch occurred in this object [125]. The authors used X-ray data from several satellites (NICER, NuSTAR, Chandra, XMM-Newton). The glitch is one of the strongest among all observed from magnetars. Relations between the glitch and the FRB-like burst were not clear. It was suggested [125] that glitches are related to active periods of magnetars, as after a glitch the magnetic field of a NS is significantly modified due to crust movements. However, a strong radio burst cannot be emitted soon after a glitch due to an abundance of charged particles in the magnetosphere. Therefore, FRB-like bursts might appear few days after glitches (but not much more, as the activity is decreasing, and necessary conditions for a fast radio burst emission are not fulfilled any more).

In October 2022, SGR 1935+2154 showed another period of activity with two FRB-like bursts accompanied by X/gamma-ray flares [126–131]. During this period of activity, glitches were not reported. However, for the period of activity in October 2020, when three FRB-like bursts were detected (without high energy counterparts), Ref. [132] there was an observations of a rapid-spin frequency variation. In this case, an anti-glitch was detected before radio bursts [133].

How glitches and anti-glitches are related to FRB-like bursts is unclear, but if this is figured out, then we can have an additional tool to study glitch/anti-glitch activity for a large sample of extragalactic magnetars.

Many other applications of FRBs in different areas of astrophysics are waiting for us in the near future. With tens of thousand of FRBs, many hundreds of which for precise redshifts will be independently measured, it is possible to perform 3D-mapping of the space medium, from the galactic ISM up to cosmological scales. The discovery of counterparts at other wavelengths, on top of all other applications, will make it possible to test Lorentz invariance at a new level of precision.

No doubt, in the next few years, we will have more galactic sources of FRBs and sources in nearby galaxies at distances $\lesssim$ a few tens of Mpc for which observations of counterparts are possible. This might help to understand the mechanism of emission and solve several other puzzles related to the physical conditions which lead to such bursts.

Proliferation of high-cadence wide-angle surveys, especially in optics (e.g., Vera C. Rubin Observatory—LSST) and X-rays, would greatly increase chances for simultaneous observations of nearby FRBs at different wavelengths [134,135]. Furthermore, high sensitivity of the next generation gravitational wave observatories (Einstein telescope, cosmic explorer) would possibly open a new area of FRB multi-messenger observations.

To summarize, in the following years studies of FRBs might open many possibilities to look deeper into the physics of NSs.

**Author Contributions:** The authors contributed equally to this review. All authors have read and agreed to the published version of the manuscript.

**Funding:** S.P. acknowledges support from the Simons Foundation which made possible his visit to the ICTP. The work of M.P. was supported by the Ministry of Science and Higher Education of Russian Federation under the contract 075-15-2020-778 in the framework of the Large Scientific Projects program within the national project "Science".

**Data Availability Statement:** No original data is presented in the review. All comments and questions might be addressed to the corresponding author.

**Acknowledgments:** We thank the referees for useful comments and suggestions. The authors actively used the NASA ADS database while preparing this review.

**Conflicts of Interest:** The authors declare no conflict of interest.

## Abbreviations

The following abbreviations are used in this manuscript:

| | |
|---|---|
| AIC | Accretion-induced collapse |
| BH | Black hole |
| CBM | Circumburst medium |
| CCSN | Core-collapse supernovae |
| CMB | Cosmic microwave background |
| DM | Dispersion measure |
| FRB | Fast radio burst |
| GR | General relativity |
| GRB | Gamma-ray burst |
| IGM | Intergalactic medium |
| IGMF | Intergalactic magnetic fields |
| ISM | Interstellar medium |
| MW | Milky Way |
| NS | Neutron star |
| PSR | Radio pulsar |
| PWN | Pulsar wind nebula |
| RM | Rotation measure |
| SGR | Soft gamma-ray repeater |
| SNR | Supernova remnant |
| WD | White dwarf |

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
