# Peer review of "Future of Neutron Star Studies with Fast Radio Bursts"

_2571-712X, doi:10.3390/particles6010025_

Round 1

Reviewer 1 Report

I have read with interest the review paper titled "Future of neutron star studies with fast radio bursts". The manuscript contains a brief discussion of several promising topics which can be developed further with help of FRB studies. These topics are quite diverse, from magnetar birth to the violation of special relativity. The manuscript is certainly very interesting and should be published. However, only after a major revision. Despite its interesting content, the manuscript looks simply unfinished and somewhat unsuitable for review. To see a clear proof of this statement one can look at the math and language of the manuscripts: many of the equations are simply wrong (contains typos and wrong coefficients) and the text is full of elementary-level language mistakes. Here are some further details on that.
* Equations
Authors need to make a significant effort in finding and correcting all the typos in the equations. Also, consider improving the consistency of the notations. Here are some examples that were spotted by the referee
- Eq. 1 uses two different notations, \nu and f, for frequency. Moreover, later in Eq. 12 "f" denotes the baryon fraction. Probably readers should profit from somewhat more consistent notations.
- Eq. 2: DM is a dimension unit, probably one should use (similarly to the frequency factors) \left(\frac{\rm DM}{\rm pc\,cm^{-3}}\right).
- Eq. 2: power index is wrong, it should be \left\frac{f}{1\rm\, GHz}\right)^{-2}
- After Eq.2: The meaning of "usually specific ... units are used" is unclear: what does it mean usually?
- Eq. 4: it seems that the coefficient 812 is wrong by three orders of magnitude (or the magnetic field is in mG units). Again a problem with dimensions. The text below ("Here $n_e$ is measured in $\rm cm^{-3}$...") does not solve the problem, and strictly speaking, it is simply wrong: in Eq. 4 notation $n_e$, $B_\|$, and $l$ correspond to two different quantities each. In the middle part of the equation, these are dimensional physical quantities and in the right part, these are normalized parameters.
- Eq. 12: $\Omega_b$ is not defined
- In Sec.4 magnetar period is $p$ and in Sec.5 is $P$
- One confuses unit acronyms (such as "sec") and unit symbols (such as "s"). This creates quite an impression of a little inaccurately written text, that may prevent readers from its proper evaluation. A small consistency check should alleviate this issue
* English
The text contains a large number of language mistakes and typos. The authors should make an effort for correcting the most obvious, such as
- confusing "few" and "a few"
- the wrong usage of "allow". It should be either "allow SB to INF" or "allow GER"
- confusing "this" and "these"
- plural and single forms of nouns
- etc

The referee believes that the authors should take an effort of revising the manuscript thoroughly and re-submit it once it is ready for review. Also, the authors should consider introducing some sketches that may help readers to see the discussed points easier.

Several more scientific issues are listed below. In general, the manuscript makes a little "pop-science" impression as it seems that it lacks some order-of-magnitude estimates in some places and further technical details in others. Please have a critical look and consider adding some further detail.

- it is unclear on what bases the authors had ruled out the possibility that FRBs can be generated also in other types of sources. For example, in https://ui.adsabs.harvard.edu/abs/2022AAS...24042506S/abstract one suggests that FRBs can be generated in X-ray binaries. If this scenario is correct, does the discussion presented in the manuscript remain unchanged? Probably a less biased discussion of existing scenarios is required.

- In the part on "properties of the surrounding medium" (Sec. 3), it is unclear how one can eliminate the host contribution, and most critically the CMB medium impact. The study of CMB medium with repeaters seems to be a very interesting and reliable case. However, it is unclear how one can get any conclusive measurement for IGM. This should be further elaborated on and explained in the manuscript. Moreover, there are no estimates for DM and RM expected from different factors. Such order-of-magnitude estimates may help readers to understand the potential of FRBs for these studies better.

- In Sec.4: what is the variability time-scale for the "magnetospheric emission mechanism"? Can one be more specific here? In the present form, the discussion is too much "hand waving"

- In Sec.5, Eq. 5: it is unclear why the simplified magneto-dipole formula should give any meaningful estimate. Could you provide some arguments?

- A more general issue in Sec. 5: Why the characteristic age is of any relevance? It seems that FRBs are associated with young magnetars and for young pulsars, the characteristic age cannot be a meaningful estimate of the system age.

- In Sec. 6: What does it mean "typical for orbital periods of binary systems"? Does it mean that the majority of binary systems with NS have this period or that systems with this orbital period are suitable for generating FRBs?

- Does one talk about 10-100 day rotation periods of a magnetar? Does one expect a relativistic outflow from such a magnetar?

Author Response

We thank the referee for detailed comments and criticism.
Our replies are given in the attached file.

Reviewer 2 Report

This paper presents an extensive review of the literature on Fast Radio Bursts, with particular attention paid to models in which they are produced by neutron stars (sometimes called "magnetars", the hyper-magnetic neutron stars believed to the the sources of Soft Gamma Repeaters).  As stated in the Abstract and Data Availability Statement, it is a review article rather than a presentation of new work.  As such, it will be a valuable resource, describing the present state of knowledge, although the expected forthcoming flood of new FRB data will make it obsolete, as stated in the Abstract. 

This paper should be published after attention is paid to the following minor points:

There may not be a consensus that the sources of all FRB are neutron stars.  This model suffers from the absence of rotational periodicity in repeating FRB; see arXiv:2207.13241 for discussion and alternatives.

Sec. 4 para. 2: The burst intensity history shown in Ref. 50 shows that the components are not separated by integer multiples of 0.217.  Although the Fourier Transform appears to indicate this, the intensity history shows several components unevenly spaced in time.

Sec. 4 para. 5: If identification of the quasiperiodicity is "not very significant", then there is no evidence it is real and nothing can be inferred.

Sec. 6: A quite different model of long-term periodicity was presented by Katz in MNRAS 516, L58 (2022); this model also attempts to explain the jitter about exact periodicity.

Author Response

We thank the referee for detailes comments.
All of them are taken into account.
Details can be found in the attached file.

Reviewer 3 Report

I think the authors have done a brilliant job at providing a review on the possible studies on neutron stars using data from fast radio bursts. We feel that the review is written very coherently and appropriate citations have been made wherever relevant. In our opinion, the article can be published in its current form without any modifications.

Regards,

The Reviewers

Author Response

We than the referee for a positive report.

Round 2

Reviewer 1 Report

The referee is satisfied with how one had accounted for the suggestions. Just a small point to consider: in DM calculations one assumes that l is measured in pc and RM in kpc. This could cause some confusion (although the units are introduced in the text).